# MMMT-IF: A CHALLENGING MULTIMODAL MULTI-TURN INSTRUCTION FOLLOWING BENCHMARK

## ABSTRACT

Evaluating instruction following capabilities for multimodal, multi-turn dialogue is challenging. With potentially multiple instructions in the input model context, the task is time-consuming for human raters and we show LLM based judges are biased towards answers from the same model. We propose MMMT-IF, an image based multi-turn Q&A evaluation set with added global instructions between questions, constraining the answer format. This challenges models to retrieve instructions dispersed across long dialogues and reason under instruction constraints. All instructions are objectively verifiable through code execution. We introduce the Programmatic Instruction Following (PIF) metric to measure the fraction of the instructions that are correctly followed while performing a reasoning task. The PIF-N-K set of metrics further evaluates robustness by measuring the fraction of samples in a corpus where, for each sample, at least K out of N generated model responses achieve a PIF score of one. The PIF metric aligns with human instruction following ratings, showing 60 percent correlation. Experiments show Gemini 1.5 Pro, GPT-4o, and Claude 3.5 Sonnet, have a PIF metric that drops from 0.81 on average at turn 1 across the models, to 0.64 at turn 20. Across all turns, when each response is repeated 4 times (PIF-4-4), GPT-4o and Gemini successfully follow all instructions only 11% of the time. When all the instructions are also appended to the end of the model input context, the PIF metric improves by 22.3 points on average, showing that the challenge with the task lies not only in following the instructions, but also in retrieving the instructions spread out in the model context. We plan to open source the MMMT-IF dataset and metric computation code.

## 1 INTRODUCTION

Despite the significant success of Large Foundation Models (LFMs) (Team et al., 2024), (Open AI, 2024), (Anthropic, 2024), (OpenAI et al., 2024) instruction following is still a challenging task (Zhou et al., 2023a). This challenge becomes more pronounced when there are multiple instructions spread out over several turns in a chat setting between a user and a LFM, where the model needs to reason over various turns of the conversation. While there are several instruction following evaluation datasets, for example (Zhou et al., 2023a), (Zhang et al., 2024), these evaluations are usually single turn and most often use text input. Another key challenge is developing objective evaluation criteria for instruction following. In collecting human annotated reference answers for our evaluation dataset, annotators reported that, at each answer turn, rewriting the answer to follow all given instructions took 10 minutes on average, highlighting that human evaluation is time intensive. Recent developments have suggested using LLMs as judges of answer quality, but we found that there was a bias in the LLM judge to favor responses coming from the same model.

A new development has been to create tasks where model answers can be programmatically checked, in the domains of coding (Yang et al., 2023), data science (Huang et al., 2022), and text (Dong et al., 2024), ensuring an objective evaluation. Among these, (Dong et al., 2024) also focus on instruction following, but only in the single turn, single modality setting. Current chat use cases are often multimodal and multi-turn, showing the need for objective instruction following evaluation datasets in this domain.

To address some of these limitations, we propose an instruction following benchmark, MMMT-IF, along with new metrics for multi-modal multi-turn dialogue. Our proposal extends the MMDU evaluation dataset (Liu et al., 2024c), a multi-modal, multi turn chat task with independent question turns. The extension adds instructions to constrain the answer format in between questions in the dialogue. All instructions are chosen so that the correctness of a response can be verified through code execution, enabling an unbiased and automated evaluation of instruction adherence. The instructions are global within a chat, meaning that all instructions from previous turns needs to be followed for future turns. Each instruction is chosen from separate categories (for example, one category dictates the start character for answer sentences, and another category dictates the end character for answer sentences), all the categories are independent from each other. Each category has either 2 or 3 instruction options. Before each question, with probability $1 - \frac{\# \text{ Instruction given so far}}{6}$ another instruction is added, uniformly at random chosen from a category (in total there are 6 categories) not yet added. This challenges the models as the task requires long context reasoning and retrieval of the instructions from different chat turns, creating a dataset that not only measures single turn instruction following performance, but also how well a model can follow multiple instructions given throughout a conversation, a common chat use case. The task is not particularly challenging for human raters, who follow on average 94% of given instructions at each turn when writing reference answers for the MMMT-IF evaluation dataset.

We develop two metrics to measure instruction following capabilities of different models: Programmatic Instruction Following (PIF), the fraction of given instructions in the chat that are followed at a certain turn, and (PIF-N-K), to stress test the ability of the models to consistently generate responses that follow all the instructions. To compute the PIF-N-K metric at a turn, we generate N responses, and the PIF-N-K metric is the fraction of the responses where at least K of the response candidates at a given turn follow all instructions, i.e. has PIF metric of 1.

We show that the evaluation suite is challenging for the models with evaluate it on: Gemini 1.5 Pro (Team et al., 2024), Claude 3.5 Sonnet (Anthropic, 2024) and GPT-4o (Open AI, 2024), with a significant loss in performance both over multiple turns and over multiple given instructions, as measured by the PIF metric. The average PIF across the models at turn 1 is high at 0.81, while at turn 20, it declines to 0.64. We develop a more nuanced measure for the model performance by considering the empirical CDF of the PIF score at each question turn. Interestingly, the empirical CDF for Sonnet 3.5 is stochastically dominated by the empirical CDF for Gemini 1.5 Pro for all question turns. This means that not only is the average PIF score better for Sonnet at each question turn, it's also true that $P(\text{PIF}_{\text{Sonnet}}(X, Y) > x | \text{turn} = i) \geq P(\text{PIF}_{\text{Gemini}}(X, Y) \geq x | \text{turn} = i)$ for all $x \in [0, 1]$, and for all turns i, for any model input context X and model response Y in the evaluation set at turn i, and $P$ is the empirical measure from all samples in the MMMT-IF evaluation set.

A similar pattern is seen when conditioned on the number of given instructions. Conditional on having given 6 instructions, the best model in our benchmark, Sonnet 3.5 has a PIF score of 0.74, and Gemini 1.5 Pro has a PIF score of only 0.4. This is in stark contrast to the PIF metric conditional on 1 instruction given, where Gemini 1.5 Pro has an average PIF score of 0.68 and Sonnet 3.5 has an average PIF score of 1 on the evaluation dataset.

For the PIF-4-K metric, the PIF-4-4 metric is only 11% for both Gemini 1.5 Pro and GPT-4o, and 28% for Claude 3.5 Sonnet, showing that all models fail to robustly follow all given instructions correctly.

We show that a significant part of the challenge with the evaluation set is not following the instructions, but rather retrieving the instructions from the model context and then reasoning over the instructions. When all instructions are added in the end of the model input context in addition to the model context, the average PIF increased 22.3 points across all models, with Gemini 1.5 Pro improving from 0.473 to 0.739, GPT-4o from 0.647 to 0.856, and Sonnet from 0.771 to 0.974, highlighting that in addition to following the instructions, retrieving the instructions from the input model context remains challenging. This highlights similarities with tasks such as multiple needles in a haystack, where the needles are instructions that needs to be reasoned over. Furthermore, our most challenging metric, the PIF-4-4 metric, showed an average improvement of 27 points, from an average of 0.16 across all models to an average of 0.43 when all given instructions were added in the end of the input model context.

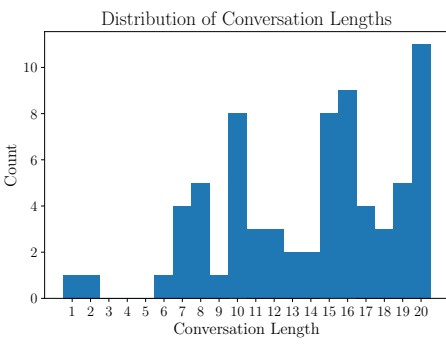

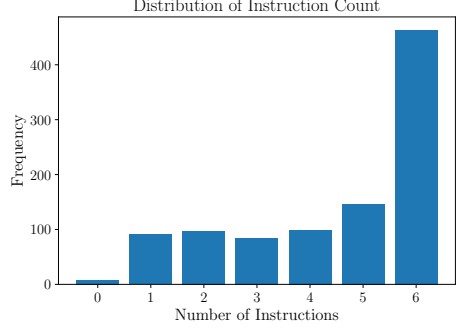

Figure 1: The number of turns of all chats in the evaluation dataset.

Figure 2: For all 990 turns, the distribution of the number of instructions that were given so far in the chat.

Further, we conducted a human study to rate the instruction following capabilities at each turn, and found out that annotators' ratings have a correlation of 60% with the proposed PIF metric.

Finally, we combine the PIF metric with an automatic LLM based evaluation of 8 criteria: accuracy, instruction following, image understanding, creativity, overall score, richness, logical coherence, and visual perception, each scored from 1 to 10. We use a weighted sum, where the PIF metric gets assigned a weight of 20 (as it's on a 0 to 1 scale) and the rest of the criteria gets assigned weight 1. We show that the resulting preference score combining the PIF metric and the Gemini 1.5 Pro based LLM judge produces preferences for overall chat quality that have an average correlation of 16% with human preference scores.

To summarize, our main contributions in this work are:

1. We propose a methodology to extend multi-modal multi-turn chat datasets to measure instruction following, implemented on the MMDU dataset.

2. We propose 2 metrics, PIF and PIF-N-K, to measure, through program execution, the effectiveness for models to follow instruction, as well as their robustness in correctly following all given instructions.

3. Through a careful analysis, we uncover a significant PIF performance degradation for all the models (Gemini 1.5 Pro, GPT-4o and Claude 3.5 Sonnet) as the number of given instructions increases.

4. We show that the main difficulty is not following the given instructions, but rather retrieving the instructions from the input model context and reasoning over them.

## 2 DATASET

This section describes the MMMT-IF evaluation dataset, as well as the human data we collect to create reference answers and preference ratings.

### 2.1 INSTRUCTION FOLLOWING EXTENSION

The instruction following extension of the MMDU dataset, to create the MMMT-IF evaluation set was described in the Introduction 1. Note that the extension makes the task also require more long context abilities in the models, as instructions needs to be retrieved from multiple parts of the input model context. Table 1 shows several statistics about the MMMT-IF evaluation set. We note that the evaluation set has 990 turn. A partial set of all the instructions and the categories is shown in Table 2. The full set of all instructions is available in Table 6 in the Appendix.

From Figure 1 we observe that that most conversations are at least 10 turns, and none are more than 20 turns.

Table 1: Descriptive Statistics of the MMMT-IF dataset

| Quantity | Value |
|---|---|
| # Chat turns | 990 |
| # Chats | 71 |
| # Images per chat | 2 - 5 |
| # Turns per chat | 1 - 20 |
| # Instructions per chat | 1 - 6 |

Table 2: Partial set of Instructions in the MMMT-IF dataset

| Name | Category | Instruction |
|---|---|---|
| Response length short | Response length | Instruction: Make all the following responses no more than 4 sentences. |
| Sentence start char S | Sentence start char | Instruction: Start every sentence with the letter (S). |
| Favorite word like | Favorite word | Instruction: Use the word 'like' at least once in all future responses. |
| Sentence length long | Sentence length | Instruction: Only use responses to questions where each sentence in the |

Each chat includes a maximum of 6 instructions, which are given before a turn with a probability inversely proportional to the number of instructions already provided. As a result, most chats will receive 6 instructions between turns 6 and 10. Given an average chat length of 14, this means that 6 instructions will be the most common number received across all turns, as shown in Figure 2. This increases the task's difficulty, as turns with more instructions are harder to satisfy completely.

## 2.2 HUMAN WRITTEN REFERENCE LABELS

We collect human labels for a reference response that both answers the questions correctly and follows all the constraints from the given instructions. In addition, the human annotators were asked to rate the answer accuracy from 1 to 10, the instruction following accuracy from 1 to 10 and give a pairwise preference score between each of the models (Gemini 1.5 Pro, GPT-4o, and Claude 3.5 Sonnet) in our evaluation set. The full set of instructions given to the human annotators is in the Appendix G.

## 2.3 DATA FILTERING

The initial evaluation set, had a total of 1342 turns, from 98 chats, the data was filtered down to 990 turns, with 71 full chats, based on the following criteria:

1. Removing chats where some image is corrupted: 23 chats

2. Removing chats with more than 5 images: 3 chats

3. Removing chats containing skipped turns due to model error or content filters: 1 chat

4. Truncating chats to have a maximum length of 20 turns.

## 3 EVALUATION METRICS

This section introduces the PIF and PIF-N-K metrics, and provides a rationale for their use.

## 3.1 PROGRAMMATIC INSTRUCTION FOLLOWING (PIF) METRIC

At each question turn, in a chat, either an instruction is added or not, with up to six instructions in each chat. Please refer to the Introduction 1 for details of how the instructions were added. Given model input context $X$, and model response $Y$, we can define the PIF metric for that response to be

$$\text{PIF}(X, Y) = \frac{\text{\# Instructions given in input context } X \text{ that are followed in response } Y}{\text{\# Instructions given in input context } X} \quad (1)$$

Note that the PIF considers whether the response follows all given instructions in previous turns, not just the instruction given at the current turn. The PIF metric does not take into account if the question was answered correctly, but rather, it focuses on if the instructions given to constrain the answer were followed.

Given a dataset $\{X_i, Y_i\}_{i=1}^{M}$ we overload the notation and define the corpus level (mean) PIF score as

$$\text{PIF}(\{X_i, Y_i\}_{i=1}^{M}) = \frac{1}{M} \sum_{i=1}^{M} \text{PIF}(X_i, Y_i) \quad (2)$$

For our evaluation set, we have M chats, and chat $m \in \{1, \ldots, M\}$ have $N_m$ turns. This gives us our evaluation set: $D = \{(X_{i,j}, Y_{i,j})\}_{i=1,j=1}^{M,N_i}$, where $X_{i,j}$ is the input model context for chat i at turn j, and $Y_{i,j}$ is the model response for chat i at turn j. The corpus level Programmatic Instruction Following Score conditioned on turn j, is given by

$$\text{PIF}(D | \text{turn} = j) = \text{PIF}(\{(X_{i,j}, Y_{i,j})\}_{i=1}^{M}), \quad (3)$$

where chats with less than j turns are excluded. It will be clear from the context whether we refer to the corpus or sample PIF metric.

The PIF metric captures the following aspects:

1. The ability for a model to retrieve several pieces of information from different parts of an input text context and reason over them
2. The ability for the model to follow objective instructions

Of these, we think the most important is the first, as this is a very common scenario for real use-cases, and it's a feature that single-turn based metrics are not capturing as well.

## 3.2 CONSISTENCY METRICS

In addition to having a high average score, we want models to consistently produce the same high quality results. We develop a metric to capture this intuition.

We propose a metric where for each turn N responses are sampled, and PIF-N-K will then denote the fraction of samples where at least $K$ samples have PIF score 1.

Thus the sample level PIF-N-K, for input model context $X$, and sampled responses $Y_1, \ldots Y_N$ is

$$\text{PIF-N-K}(X, Y_1, \ldots, Y_N) = \begin{cases} 1, & \text{if } \sum_{i=1}^{N} \mathbf{1}_{\text{PIF}(X, Y_i) = 1} \geq K \\ 0, & \text{otherwise} \end{cases} \quad (4)$$

The intuition is that we want to measure how consistently the models can follow all the instructions correctly. We overload notation and define the corpus level (mean) PIF-N-K for a dataset with M turns, $D = \{X_i, Y_{1,i}, \ldots, Y_{N,i}\}_{i=1}^{M}$ as

$$\text{PIF-N-K}(D) = \frac{\sum_{j=1}^{M} \text{PIF-N-K}(X_i, Y_{1,i}, \ldots, Y_{N,i})}{M}.$$

With this definition it holds that, for any dataset $D$,

$$\text{PIF-N-i}(D) \leq \text{PIF-N-j}(D) \quad (5)$$

when $i > j$.

## 4 EVALUATED MODELS

This section describes the models evaluated, and provides an analysis of the answer lengths of the models.

### 4.1 MODEL ENDPOINTS

We access Gemini 1.5 Pro (abbreviated as Gemini) through the Vertex AI API, using the following model version: 'Gemini-1.5-pro-preview-0514'. We access Claude 3.5 Sonnet (abbreviated as Sonnet) through the Anthropic Vertex API, with the model version 'claude-3-5-sonnet@20240620'. We access GPT-4o from the OpenAI API with the model version 'gpt-4o-2024-05-13'. The hyperparameters for all models are the default settings. The default temperature for all models in 1. The safety filters for all models are the default settings. We don't see questions that are marked as unsafe with the default setting for the models.

### 4.2 CONTEXT LENGTHS

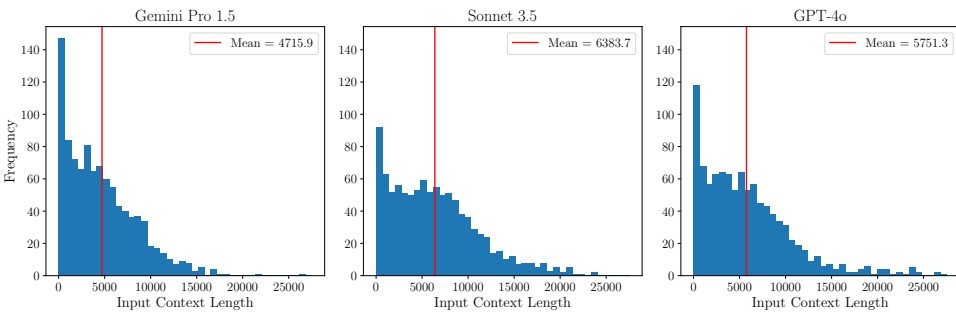

Figure 3: The distribution of the input context lengths for Gemini 1.5 Pro, Claude 3.5 Sonnet and GPT-4o, in the evaluation dataset, along with the mean input context length in characters.

Figure 3 shows that the mean input context length for Gemini 1.5 Pro is the smallest, as the input context is made up from the questions and model outputs in the previous turns, and the average output generated is shortest by Gemini 1.5 Pro. This does not take into account the images that are inputted at the beginning of each chat. It also shows that the average input context is rather long, thus requiring long context reasoning.

## 5 EVALUATION RESULTS

The section describes the results from the evaluation experiments, starting with results for the PIF metric, then considering similarities with the needle in a haystack experiment, results for the PIF-N-K metric, before finally considering human and LLM-as-a-judge evaluation results.

### 5.1 PIF METRIC

Figure 4 shows the PIF conditional on question turn. We note that, as expected, the PIF metric decreases with the question turn. The 95% confidence bounds for the PIF metric are done on a per-turn basis, using a Bernoulli confidence interval approximation. This gives conservative confidence bounds as the Bernoulli distribution is the distribution that for a given mean maximizes the variance among all distributions on [0,1].

Figure 6 shows the empirical cumulative distribution function for the PIF metric. The interpretation of the left graph in Figure 6 is that at turn 2, the programmatic instruction following score can be 0, 0.5, or 1. For Gemini 1.5 Pro, it's 0 with probabilty 18%, while for GPT-4o it's zero with probability around 10%. The probability that the programmatic instruction following score is less than 1 (i.e 0.5 or 0) is around 35% for GPT-4o, 52% for Gemini and 10% for Sonnet. In the right plot of Figure 6,

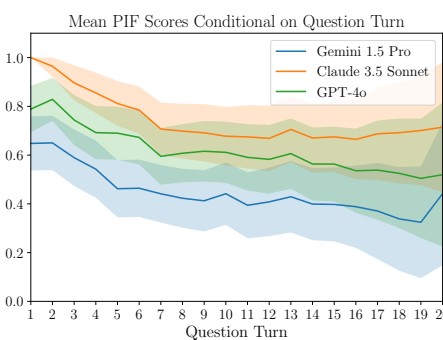
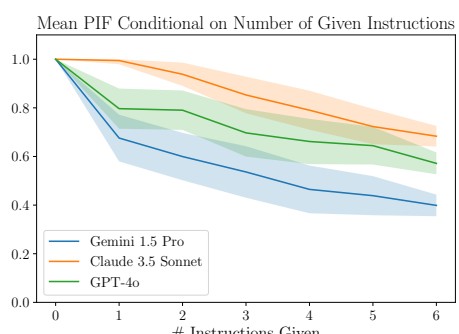

Figure 4: The image above shows the PIF metric conditioned on the question turn with 95% confidence intervals. For a fixed turn $i$, the mean is taken across all chats at with at least $i$ turns.

Figure 5: The mean programmatic instruction following score conditional on the number of instructions given in the chat so far. The metric defaults to 1 if no instruction has been given.

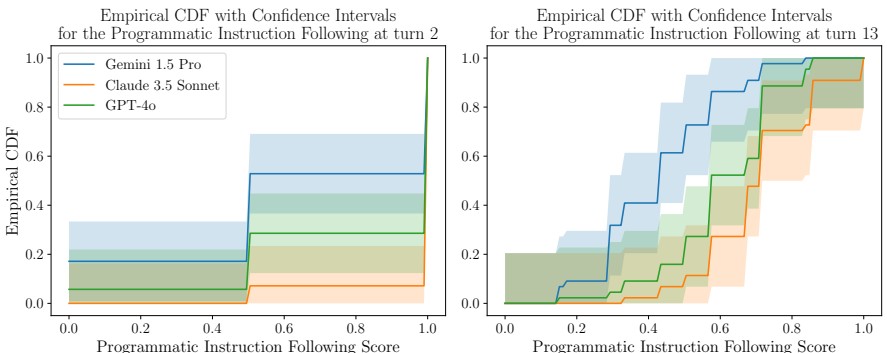

Figure 6: The empirical CDF of the PIF metric conditional on question turn, with confidence intervals.

we see the empirical CDF PIF of Gemini stochastically dominating that of Sonnet. Interestingly, the empirical CDF for Sonnet 3.5 is stochastically dominated by the empirical CDF for Gemini 1.5 Pro for all question turns 1-20. This means that not only is the average PIF score better for Sonnet at each question turn, it's also true that $P(\text{PIF}_{\text{Sonnet}}(X, Y) > x | \text{turn} = i) \geq P(\text{PIF}_{\text{Gemini}}(X, Y) \geq x | \text{turn} = i)$ for all $x \in [0, 1]$, and for all turns i, for any model input context X and model response Y in the evaluation set at turn i, and $P$ is the empirical measure from all samples in the MMMT-IF evaluation set.

From Figure 5 we see that the scores decrease with the number of given instructions, as it's harder for the models to follow multiple instructions at the same time. Also note that Gemini 1.5 Pro has a significantly lower score for high number of instructions compared with Sonnet and GPT-4o, highlighting an area for improvement. Finally, note that the programmatic instruction following metrics is automatically evaluated by code execution, which significantly increases the reliability of the shown results. Also note that Gemini has strong performance when only given a single instruction. The 95% confidence intervals are computed with a Bernoulli approximation.

## 5.2 NEEDLES IN A HAYSTACK?

This experimental setup has several similarities and differences with a needle in a haystack experiment. In our proposed set up, the complex reasoning across the needles (given instructions) is important, in addition to the retrieval of the needles. To understand the impact of where in the input model context the instructions are located, we run the following ablation: In addition to having in-

Table 3: Programmatic Instruction following on MMMT-IF Evaluation Set

| Metric | Gemini 1.5 Pro | GPT-4o | Sonnet 3.5 |
|---|---|---|---|
| Programmatic Instruction Following (PIF) | 0.473 | 0.647 | 0.771 |
| PIF with all instructions added at end of input prompt | 0.739 | 0.856 | 0.974 |

structions throughout the input context, we add all given instructions at the end of the input model context. Table 3 shows the results, where we see that the corpus level PIF increased 22.3 points on average across all models, highlighting that in addition to following the instructions, retrieving the instructions from the input model context remains challenging.

On a practical level, this suggests an easy method to improve instruction following capabilities in multi-turn chat: find all the instructions and add them to the end of the input model context.

The first row of Table 3 also highlights statistically significant differences in the corpus level (all 990 turns) PIF metric between the evaluated models. We see that the programmatic instruction following score is best for Sonnet, and Gemini has the weakest performance. Using the (non parametric) Wilcoxon Signed Rank test, we reject the hypotheses $H_0 : P(\text{PIF}_{\text{Gemini}} > \text{PIF}_{\text{Sonnet}}) >= 0.5$ with p-value smaller than $10^{-5}$. Using the Wilcoxon Signed Rank test, we also reject the hypotheses $H_0 : P(\text{PIF}_{\text{Gemini}} > \text{PIF}_{\text{GPT-4o}}) >= 0.5$ with p-value smaller than $10^{-5}$. The conclusion is the difference between the models for the mean programmatic instruction following metric is significant.

## 5.3 PIF-N-K Metric

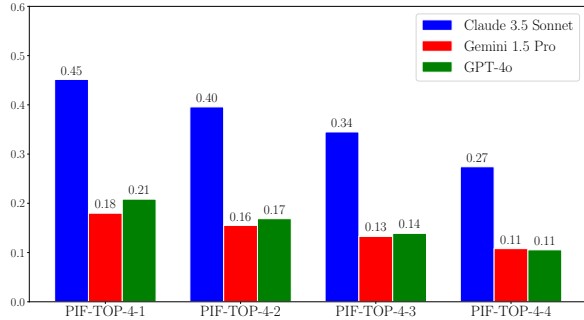

Figure 7: The corpus level PIF-4-j metrics for $j \in \{1, 2, 3, 4\}$

We now consider the results for the PIF-N-K, measuring the robustness for following all given instructions correctly. In our experiments we set $N = 4$. Figure 7 shows the results. As expected PIF-4-4, meaning the fraction of turns where all $N = 4$ sampled turn answer candidates got all the instructions correct is quite low, for both Gemini and GPT-4o it's 11%, highlighting that this is a very challenging metric with significant headroom for model improvement. However, note that also for Sonnet 3.5, the model with the strongest performance, the metric rapidly becomes more challenging as we move from PIF-4-1 to PIF-4-4. This points to a significant robustness issue with the models we have studied in this work, as if the model always had the same percentage of instructions followed in its responses, we would not see a decrease in the PIF-N-K metric.

## 5.4 Human Evaluation

As described in Section 2.2 we collect human evaluations of instruction following, chat accuracy and pairwise preferences. In Figure 8 with human evaluations, we observe that Gemini underperforms GPT-4o and Sonnet, and Sonnet and GPT-4o are broadly similar. We also find that the correlation between the human instruction rating and the PIF metric, the results are shown in Table 4. We

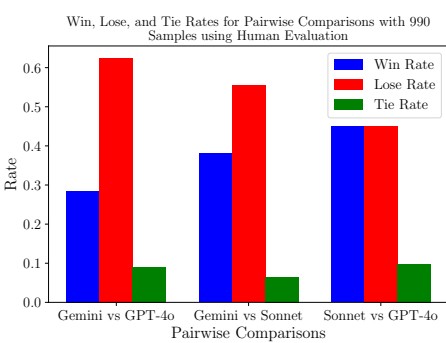

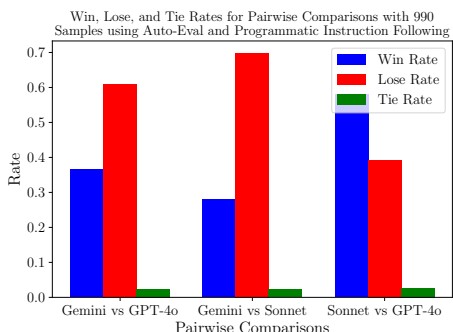

Figure 8: Win, Tie, and Loss rate using a human preference ranking for Gemini vs GPT-4o, Gemini vs Sonnet, and Sonnet vs GPT-4o.

Figure 9: Win, Tie, and Loss rate using Gemini based autorater preference ranking for Gemini vs GPT-4o, Gemini vs Sonnet, and Sonnet vs GPT-4o.

Table 4: Correlation between the PIF metric and the human rated instruction following metric for 990 samples from human raters.

| Overall Correlation | Gemini 1.5 Pro | GPT-4o | Sonnet 3.5 |
|---|---|---|---|
| 0.60 | 0.44 | 0.68 | 0.63 |

note that the average correlation across all models is high, $0.60$, indicating the usefulness of the PIF metric to capture the instruction following of the models. In Table 5 we see that the average human evaluation score for accuracy is highest for GPT-4o, highlighting that while PIF score is an important metric, there are several aspects of model performance the metric does not cover.

### 5.4.1 How hard is the task for human raters?

Starting with a reference answer from the original MMDU dataset, human raters were instructed to rewrite the responses to both be correct and to follow all the given instructions. The human raters had access to the LLM model responses, the original reference answer for the MMDU dataset, as well as a list of all instructions given in the chat, so they did not have to look at the chat history to find the instructions. The raters reported that it took on average 10 minutes to write the answer and reported that the hardest instructions to satisfy where the constrains on the sentence start word and the constrains on the sentence lengths. The programmatic instruction following scores for the human raters have an average of 0.94, significantly higher than both Gemini and GPT-4o with all instructions in the end of the input context, but actually lower than Sonnet 3.5 in the setting with all instructions added at the end of the input model context, at a mean PIF score of 0.97. This highlights that while the task is challenging, the human raters are able to complete it with great proficiency, indicating that there is headroom for models to improve. The raters reported that having access to the model answers helped speed up the rewriting process by giving inspiration to ways to follows the given constrains. The raters also noted that sometimes the original reference answers from the MMDU dataset were incorrect and had to be adjusted in addition to ensuring that all instructions were followed.

### 5.5 Autorater

### 5.5.1 Autorater Metrics

We use an LLM based autorater to rate the chats on 8 metrics: Creativity, Richness, Visual Perception, Logical Coherence, Answer Accuracy, Image Relationship Understanding, Instruction Following and Overall Quality, each on a scale from 1 to 10. The autorater is given the chat history, model response, and input and outputs a dictionary with the score for each attribute. We run experiments

Table 5: Gemini vs GPT-4o as Autorater vs human evaluation on the MMMT-IF dataset

| Judge | Gemini 1.5 Pro | GPT-4o | Human |
|---|---|---|---|
| Avg Accuracy Gemini | 6.95 | 7.36 | 6.04 |
| Avg Accuracy GPT-4o | 7.07 | 7.82 | 6.70 |
| Avg Accuracy Sonnet | 6.92 | 7.44 | 6.33 |
| Avg Instruction Following Gemini | 7.61 | 8.33 | 3.80 |
| Avg Instruction Following GPT-4o | 7.65 | 9.06 | 4.41 |
| Avg Instruction Following Sonnet | 7.81 | 9.01 | 5.32 |

with both Gemini 1.5 Pro and GPT-4o as the autorater LLM. The definition for the autorater metrics is based on the work in (Liu et al., 2024c).

### 5.5.2 PAIRWISE COMPARISONS

For each of the 8 metrics, a score in the range 1-10 is given by the auto-rater model. For the programmatic instruction following a score in the range 0 - 1 is given. We create a final response score by taking a weighted average of all the autorater scores and the programmatic instruction following score, where the programmatic instruction following score has a weight of 20, as it's scored on a more narrow range, and it's the metric that is most objective. Using this weighted average, we can compare the responses for Gemini 1.5 Pro, GPT-4o and Sonnet 3.5 for each chat turn.

From the preference scores in Figure 9 we see that Gemini significantly lags behind both Sonnet and GPT-4o on the task, and in particular against Sonnet, the win rate is only around 28%.

**Autorater bias** From Table 5 we make an interesting, but not surprising observation: With GPT-4o as judge, GPT-4o performs better, and with Gemini as a judge, Gemini has a better performance. The Gemini based autorater gets the relative order of the instruction following correct (as measured by the programmatic instruction following metric) whereas the GPT-4o judge ranks the GPT-4o as having better instruction following than Sonnet. In addition, we see that the human rater scores are in general more conservative. For the accuracy ratings, the GPT-4o judge has the same relative ranking as the human raters, which the Gemini judge does not.

## 6 CONCLUSION

In this work we proposed the MMMT-IF instruction following evaluation set for multi-modal, multi-turn dialogue, along with several challenging metrics for the current LFMs. All the metrics are objective and verifiable by code execution, ensuring an unbiased evaluation. Our analysis shows that the main difficulty of the task lies not within the instruction following, but rather to retrieve the instructions from different parts of the input context and then reason over them. We find that Gemini 1.5 Pro consistently, and to a lesser extent also GPT-4o and Claude 3.5 Sonnet, have a significant PIF metric degradation on the turns in the evaluation set with many instructions. We also find that all examined models have low PIF-N-K scores, indicating that they fail to robustly follow all given instructions correctly. We hope that the PIF and the PIF-N-K metric can serve more broadly for practitioners wishing to create other evaluation benchmarks for multimodal, multi-turn instruction following.

Possible directions for future work include creating training datasets for Reinforcement Learning from Execution Feedback (RLEF) with the PIF and PIF-N-K as reward signals. Another possible extension include creating dependent instructions, such as having instructions that modify previously given instructions. This will make the task further more challenging.

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

# A LITTERATURE REVIEW

## A.1 LONG CONTEXT RETRIEVAL

There have been several works focusing on the effect of long input context on model performance on downstream tasks, including (Liu et al., 2023), (Levy et al., 2024), and (An et al., 2023). Similar to the Lost-in-the-middle paper Liu et al. (2023), our paper examines the effect of where in the input context information is located. The results in (Levy et al., 2024) are also complementary, as both observe performance degradation with input context length increases. Our evaluation set can also be viewed as a task similar to multiple needles in the haystack (a task where several tokens needs to be retrieved from a long input context), where each needle is an instruction that the model needs to reason over.

## A.2 INSTRUCTION FOLLOWING EVALUATION

There a many instruction following evaluation benchmarks, several are, like our work, focusing on instructions related to answer constrains on a Q&A task, (Xia et al., 2024), (Zhou et al., 2023a),

(Zhang et al., 2024), (Tam et al., 2024) and (Sun et al., 2024a). Compared with these works, we focus on multiple instructions spread out over a long context, testing not only instruction following but also retrieval and complex reasoning from the input context. There have been a long range of other instruction following evaluation sets (Chen et al., 2024c), (Zhou et al., 2023b), (Adlakha et al., 2024), (Sun et al., 2024b), (Yan et al., 2024), (Jiang et al., 2024), (Chia et al., 2023), (Skopek et al., 2023), and (Qin et al., 2024), but their focus in not on multiple instructions spread out in the input context for multi-modal multi-turn chat.

### A.3 PROGRAMMATIC INSTRUCTION FOLLOWING

There have been several previous papers that use program execution to determine instruction following capability, for code (Yang et al., 2023), Data science (Huang et al., 2022), and text (Dong et al., 2024). Our work is most related to (Dong et al., 2024), but we fix a set of instructions, and instead of a single instruction use case, we focus on multiple instructions, over multiple turns of multi-modal question answering.

### A.4 MULTI-MODAL EVALUATION DATASETS

There have been several benchmarks suggested for multi-modal models, for the multi-turn chat use case, we have (Liu et al., 2024c) and (Liu et al., 2024a). However, while the datasets are multi-turn, the chat turns can be independently answered, thus making it less relevant for long context models. By introducing given at several locations throughout the chat, we introduce long range dependencies in the data needed to answer questions. Our evaluation set is an extension of the MMDU dataset (Liu et al., 2024c). Other work for evaluating multi-modal models include (Yue et al., 2024), (Liu et al., 2024b), (Srinivasan et al., 2021), (Yu et al., 2023), (Xu et al., 2023), (Chen et al., 2024b) and (Wang et al., 2024). None of these focus on multi-turn instruction following.

### A.5 LLM JUDGES

There have been several previous works on using LLMs to judge quality of other LLM responses, including (Dubois et al., 2024b), (Zheng et al., 2023), (Chen et al., 2024a), (Dubois et al., 2024a), (Zeng et al., 2024), and (Liu et al., 2024c). Compared with these works, our LLM judge approach is different in that we combine an LLM evaluation with a programmatic evaluation through a weighed sum to create final model evaluations.

## B ADDITIONAL EXPERIMENTS

### B.1 PERFORMANCE ON SPECIFIC INSTRUCTIONS

In Figure 10 the PIF score conditional on an instruction having been given is shown. We note that Gemini 1.5 Pro has a hard time following an instruction to end sentences with a question mark, and GPT-4o has some issues with following instructions related to outputting even or odd numbers in its responses. The definition of the categories are presented in Table 6 in the Appendix.

### B.2 ANALYSIS OF DATASET QUESTIONS

In Figure 12 we display the model capabilities targeted in each question turn, where the classification is done by GPT-4o. We manually reviewed the classifications to ensure they were aligned with human categorizations.

In Figure 11, we show the average response length of conditioned on the LLM capability the question targets. We see that questions classified as Creativity and Visual Comparative Analysis have longer average answer lengths compared with those classified as visual object description.

Rather then focus on the LLM capability, Figure 13 shows the distribution of the questions topics in the dataset, classified by GPT-4o. Many questions are related to flowers, plants, architecture, food and vehicles.

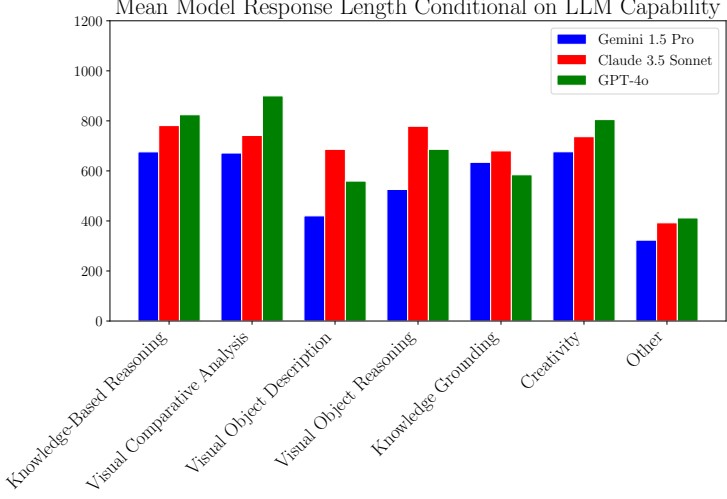

Figure 10: the mean conditional programmatic instruction following score conditioned on an instruction having been given in the chat.

Figure 11: The mean answer length conditional on the LLM capability the question most closely targets.

## B.3 PIF METRIC AND HUMAN ACCURACY SCORES

While the PIF score is an important metric for instruction following, it's also important to answer the image based questions in the MMMT-IF dataset correctly, not only following the answer constraints. Figure 14 shows a scatter plot with PIF score on the y axis and human accuracy score for each turn on the x axis. It's desirable to both have high accuracy score and high PIF score, but this is relatively uncommon as shown in the Figure, highlighting the challenge of the task. In the Figure the cluser centroids are also shown. Note that GPT-4o responses have the highest average human accuracy scores and Claude 3.5 Sonnet have the highest average PIF scores. Also note that the

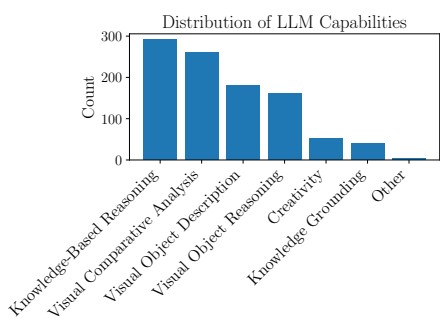

Figure 12: The distribution of LLM capabilities that the questions in the dataset targets.

Figure 13: The questions topics, as rated by an GPT-4o with access to the associated images and verified by a human for each turn in the MMMT-IF eval set.

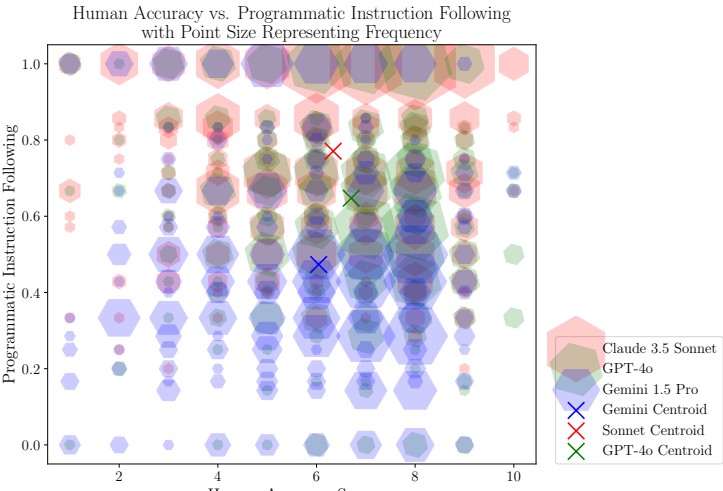

Figure 14: Scatter plot for Gemini 1.5 Pro, GPT-4o and Claude 3.5 Sonnet responses with PIF scores on the y axis and human accuracy scores on the x axis. The size of the points is proportional to the number of samples with the same PIF score and human accuracy score.

Sonnet responses have more robustly high PIF score, and the GPT-4o responses have more robustly high human accuracy scores.

### B.4 ADDITIONAL ANALYSIS OF MODEL RESPONSE LENGTH

While the PIF score is a key metric for evaluating instruction-following, it's also important that models answer the image-based questions in the MMMT-IF dataset correctly, beyond just following answer constraints. Figure 14 shows a scatter plot with PIF scores on the y-axis and human accuracy scores on the x-axis for each turn. Ideally, responses should achieve both high accuracy and high PIF scores, but this combination is relatively rare, as shown in the figure, underscoring the difficulty of the task. Cluster centroids are also highlighted. Notably, GPT-4o responses have the highest average human accuracy scores, while Claude Sonnet 3.5 responses achieve the highest average PIF scores. Additionally, Sonnet's responses show more consistent high PIF scores, whereas GPT-4o have more robustly high human accuracy scores.

Figure 15: Mean response length (in characters) conditional on question turn in the MMMT-IF evalution set.

Table 6: Full set of Instructions in the MMMT-IF dataset

| Name | Instruction |
| --- | --- |
| Response length short | Instruction: Make all the following responses no more than 4 sentences. |
| Response length long | Instruction: Make all the following responses at least 5 sentences. |
| Sentence start char S | Instruction: Start every sentence with the letter (S). |
| Sentence start char B | Instruction: Start every sentence with the letter (B). |
| Sentence end char ? | Instruction: End every sentence with a question mark (?). |
| Sentence end char ! | Instruction: End every sentence with an exclamation mark (!). |
| Favorite word like | Instruction: Use the word 'like' at least once in all future responses. |
| Favorite word itself | Instruction: Use the word 'itself' at least once in all future responses. |
| Favorite word per se | Instruction: Use the word 'per se' at least once in all future responses. |
| Sentence length short | Instruction: Only use responses to questions where each sentence in the response is at most 18 words in all future responses. |
| Sentence length long | Instruction: Only use responses to questions where each sentence in the response is at least 18 words in all future responses. |
| Number even | Instruction: Include at least one even number bigger than 5 in each of your responses. |
| Number odd | Instruction: Include at least one odd number bigger than 5 in each of your responses. |

## C  FULL SET OF INSTRUCTIONS

The full set of instructions is given in Table 6.

## D  ADDITIONAL METRICS

### D.1  PIF-IQR-N METRIC

Here we define an additional metric for robustness that focus on overall robustness rather than robustness for correctly following all instructions as in PIF-N-K. For a given input model context $X$, we sample responses $Y_1, \ldots, Y_N$, to get PIF scores $\text{PIF}(X, Y_1), \ldots, \text{PIF}(X, Y_N)$. Now we define

PIF-IQR-N as

$$\text{PIF-IQR-N}(X, Y_1, \ldots, Y_N) = \text{IQR}(\{\text{PIF}(X, Y_1), \ldots, \text{PIF}(X, Y_N)\}), \tag{6}$$

where IQR denotes the interquartile range. This measure has the property to in expectation to be independent of N.

### D.1.1 ELO COMPUTATION

Another way to use the pairwise preference ratings from the weighted sum of the autorater and the PIF metric is to compute the ELO score between the three models. We also compute an ELO score for each of the models, where the ELO is initialized at 1000. We follow the procedure in (Chiang et al., 2024).

We get that Gemini has ELO 950, GPT-4o has an ELO of 994, and Sonnet has an ELO of 1055. This ELO ranking suggests a win probability of 35% between Gemini and Sonnet, 44% between Gemini and GPT-4o and 59% between Sonnet and GPT-4o, which is broadly aligned with the observed win-rates, although Sonnet has a higher winrate compared to the human preference ratings.

## E EXAMPLE CHAT

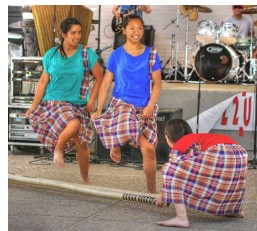
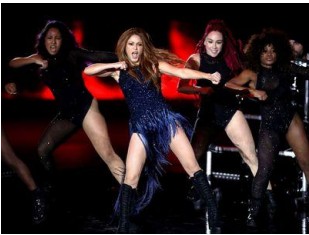

(a) Image 1 in example chat    (b) Image 2 in example chat

Figure 16: Images for the example chat

In Table 7 we show an example chat and Figure 16 shows the images corresponding to the chat. In Table 7, the column "All instructions" shows all the instructions given in previous and current chat turns. However, the model will not be directly given the data in this column, but rather needs to find the instructions in the input context, where each instructions appear before questions in the user turns. The PIF column shows the PIF score for the considered chat turn.

## F ERROR ANALYSIS

In this section we explore a few chats and the errors made. While the average performance of the PIF metric for human raters was at 0.94, the lowest observed PIF score for a chat turn was 0.4, the chat turn is shown in Table 8. Note that the word 'like' is not in the response, and that sentence 7 is both longer than 18 words and start with the letter T, hence the PIF score of 0.4. Notice that the answer looks broadly correct and it requires a careful view to spot the errors.

**Gemini repeats the same answer** One error pattern noticed for Gemini 1.5 Pro was that it's repeating the same answer to multiple questions, see row 1-5 in Table 9, which are consecutive turns in a conversation. In row 5, Gemini follows 0 out of the 4 given instructions.

**Gemini gives up** In Table 10, we show a response where gemini claims the task is impossible, when it's not, and hence fails to adhere to any of the six given instructions.

**GPT-4o ignores number instruction** In Table 11, we show several chats where GPT-4o ignores instructions related to numbers.

Table 7: Chat example

| All instructions | Turn | Question | Gemini Response | PIF |
|---|---|---|---|---|
| Instruction: Include at least one even number in the range bigger than 5 in each of your responses. | 1 | What activity are the individuals in the Image1 engaged in, which is characterized by rhythmic and fast-paced footwork? | The individuals in Image 1 are performing Tinikling, a traditional Philippine folk dance characterized by rhythmic and fast-paced footwork, often using bamboo poles. Dancers need a lot of agility to avoid getting their feet caught between the poles, especially as the pace increases to 10 or 12 beats per second. | 1 (1/1) |
| ⋮ | ⋮ | ⋮ | ⋮ | ⋮ |
| Instruction: Include at least one even number in the range bigger than 5 in each of your responses. Instruction: End every sentence with a exclamation mark (!). Instruction: Use the word 'itself' at least once in all future responses. Instruction: Only use responses to question where each sentence in the response is at most 18 words in all future responses. Instruction: Make all the following responses no more than 4 sentences. Instruction: Start every sentence with the letter (S). | 15 | Identify the possible significance of the props or lack thereof in the settings of Image1 and Image2. | Stage props can enhance the narrative or symbolism of a performance. Simple props, like the bamboo poles in Image 1, become integral to the dance itself. Conversely, the absence of props in Image 2 might emphasize the dancers' movements and stage presence. Such choices reflect deliberate artistic decisions in conveying a performance's message. | 0.5 (3/6) |

## G  HUMAN ANNOTATOR INSTRUCTIONS

In this section the instructions given to human annotators are presented. There were 8 human annotators working on the MMMT-IF dataset. The work was split so that all turns in a chat were rated and rewritten by the same person. Below are the full set of instructions given to human annotators:

**Requirement**  For each sample (total 71) based on the MMDU benchmark (MMDU: A Multi-Turn Multi-Image Dialog Understanding Benchmark and Instruction-Tuning Dataset for LVLMs), several images (up to 5) are shown and multiple questions and instructions (average 14 questions and 6 instructions be sample) related to the images are asked based on this. Answers from several models are provided for each question, following the instructions and answering the question. In total there are 990 questions (71 conversations each with an average of 14 turns). Part 1: Each question has a reference answer that should be rewritten to ensure it follows the given instructions in addition to correctly answering the question about the images. Part 2: For each of the three models and for each question, individual ratings for instruction following and answer accuracy should also be evaluated based on the metric in the appendix, both on a scale from 1 to 10. For each question, 3 model answers will be compared, Gemini, GPT-4o, and Sonnet 3.5. The second part is to provide a side by side

Table 8: The chat turn with the lowest PIF score for the human rater.

| All instructions | Question | Human Rater Answer | PIF |
|---|---|---|---|
| Instruction: Include at least one even number in the range bigger than 5 in each of your responses. Instruction: Only use responses to question where each sentence in the response is at most 18 words in all future responses. Instruction: Start every sentence with the letter (B). Instruction: Use the word 'like' at least once in all future responses. Instruction: End every sentence with a exclamation mark (!). | What could be the symbolic meanings of the food items in Image1 and Image2 in the context of their respective culinary cultures? | Both food items in Image1 and Image2 convey different storylines and sentiments within their culinary cultures! Being round and sweet, doughnuts, as depicted in Image1, often symbolize indulgence and reward! Because of their round shape, they can also denote eternity and completeness! By comparison, spaghetti and tomato sauce shown in picture 2, hold deep-rooted symbolism in Italian culture! Being central to many meals, spaghetti symbolizes unity and family cohesion for more than 6 centuries! By making pasta from scratch or preparing a family recipe the second dish speaks to continuity and heritage! The tomato sauce signifies prosperity and abundance, often representative of the lush Italian gardens and the healthful Mediterranean diet! Beyond its flavor, tomato sauce symbolizes prosperity, reflecting Italy's lush gardens, as well as a healthy Mediterranean diet! Both pasta and tomato sauce embody Italy's identity and cultural pride with connection to land, tradition, and family! | 0.4 (2/5) |

preference ranking between Gemini and GPT-4o, Gemini and Sonnet, as well as between Sonnet and GPT-4o. For each comparison, write down the winner model's name ("Gemini", "GTP-4o", or "Sonnet"), or "tie". The comparison should be based on the similarity to the rewritten reference answer from part 1.

**Relevant columns** Id: The sample number of the chat. All rows with the same id belong to the same chat sample. question: The question that the models need to answer. All instructions: All instructions that need to be followed when writing the updated reference answer. Gemini-pro-1.5 answer: Question response generated by Gemini-1.5 Pro, can be used for inspiration for the new reference answer. GPT-4o answer: The question response generated by the GPT-4o model, can be used for inspiration for the new reference answer. Sonnet-3.5 answer: Question response generated by Sonnet 3.5, can be used for inspiration for the new reference answer. Image1-image5: images related to the question on the same row. Original (no instructions) reference answer: reference answer not taking the instructions into account

**Columns to be filled out for Part 1** Rewritten reference answer: FILL OUT new rewritten reference answer that uses all given instructions. True facts can be added in order to fulfill the instructions. Optional Comments rewritten reference answer: If you have any comments about the creation of the rewritten preference answer, fill out this column. This is optional.

**Column to be filled out for Part 2** Answer Accuracy Gemini (Human Preference): FILL OUT the answer accuracy for Gemini-1.5 Pro on 1-10 scale (see the rubric in appendix) Answer Accuracy GPT-4o (Human Preference): FILL OUT the answer accuracy for Gemini-1.5 Pro on 1-10 scale (see the rubric in appendix) Answer Accuracy Sonnet 3.5 (Human Preference): FILL OUT the answer accuracy for Gemini-1.5 Pro on 1-10 scale (see the rubric in appendix) Instruction following Gemini (Human Preference): FILL OUT the instruction following skill for Gemini-1.5 Pro on 1-10 scale (see the rubric in the appendix) Instruction following GPT-4o (Human Preference): FILL OUT

the instruction following skill for Gemini-1.5 Pro on 1-10 scale (see the rubric in the appendix) Instruction following Sonnet 3.5 (Human Preference): FILL OUT the instruction following skill for Gemini-1.5 Pro on 1-10 scale (see the rubric in the appendix) For both the instruction following and the accuracy metric, the scoring should be based on comparison with the rewritten reference answer from part 1.

Preference: Gemini vs GPT-4o (Human Rating): FILL OUT, "Gemini" if the Gemini response is preferred, "GPT-4o" if the GPT-4o response is preferred. Preference: Gemini vs Sonnet (Human Rating): FILL OUT, "Gemini" if the Gemini response is preferred, "Sonnet" if the Sonnet response is preferred Preference: Sonnet vs GPT-4o (Human Rating) : FILL OUT, "Sonnet" if the Sonnet response is preferred over the GPT-4o response, "GPT-4o" if the GPT-4o response is preferred. Comments Preference Rating: If you have any comments about your entry, you can optionally fill out this column. For each of the preference ratings, a score of "tie" can also be given if the responses are almost identical. However, only use the "tie" category when strictly necessary.

**Metrics**  Answer Accuracy

- Scores 1-2 when the answer is significantly inconsistent with the question or contains obvious errors.
- Scores 3-4 when the answer is partially correct but contains some errors or is incomplete, significantly worse accuracy compared to the rewritten reference answer.
- Scores 5-6 when the answer is basically correct but lacks details or is not sufficiently detailed, the accuracy is worse than the reference answer.
- Scores 7-8 when the answer is accurate and detailed, fully corresponding to the question, on par with the reference answer.
- Scores 9-10 when the answer is not only accurate and detailed but also provides additional useful information, exceeding the rewritten reference answer.

Instruction Following

- Scores 1-2 when the answer is completely ignoring most or all of the previously given instructions.
- Scores 3-4 when several of the instructions are followed but some are not followed, significantly worse than the rewritten reference answer.
- Scores 5-6 when most of the instructions are correctly followed, but there are some errors, worse quality than the rewritten reference answer.
- Scores 7-8 when all instructions except perhaps 1 is followed in a good way, on par with the rewritten reference answer.
- Scores 9-10 when all instructions are followed in a clear and insightful way, exceeding the rewritten reference answer.

Table 9: A chat where Gemini is repeating the same answer

| All instructions | Question | Answer | PIF |
|---|---|---|---|
| Instruction: Start every sentence with the letter (S). | What type of environment is depicted in the Image1? | Swamps are the typical environment in image 1. | 1 (1/1) |
| Instruction: Start every sentence with the letter (S). | Based on the Image1, how are the cattail plants predominantly arranged or positioned in their natural habitat? | Swamps are the typical environment in image 1. Spatially, they appear to be clustered together in dense stands. | 1 (1/1) |
| Instruction: Start every sentence with the letter (S). Instruction: Only use responses to question where each sentence in the response is at least 18 words in all future responses. | Considering the Image1, what practical uses do the cattail leaves serve in various cultures? | Swamps are the typical environment in image 1. Spatially, they appear to be clustered together in dense stands. Serving a multitude of purposes, the leaves of the cattail plant have been ingeniously incorporated into diverse aspects of life in various cultures throughout history. Specifically, the leaves are recognized for their strength and pliability, making them ideal for weaving into mats, baskets, and even temporary shelters, providing both comfort and protection from the elements. | 0.5 (1/2) |
| Instruction: Start every sentence with the letter (S). Instruction: Only use responses to question where each sentence in the response is at least 18 words in all future responses. Instruction: Include at least one even number in the range bigger than 5 in each of your responses. | What type of flowers are depicted in the Image2, and what might be their historical significance in floral arrangements? | Swamps are the typical environment in image 1. Spatially, they appear to be clustered together in dense stands. Serving a multitude of purposes, the leaves of the cattail plant have been ingeniously incorporated into diverse aspects of life in various cultures throughout history. [...]. The exquisite flowers depicted in Image 2 are Cattleya orchids, renowned for their vibrant hues and intricate blooms, captivating hearts for centuries. [...] | 0.33 (1/3) |
| Instruction: Start every sentence with the letter (S). Instruction: Only use responses to question where each sentence in the response is at least 18 words in all future responses. Instruction: Include at least one even number in the range bigger than 5 in each of your responses. Instruction: Make all the following responses no more than 4 sentences. | Describe the primary colors and physical characteristics of the flowers in the Image2. | Swamps are the typical environment in image 1. Spatially, they appear to be clustered together in dense stands. Serving a multitude of purposes, the leaves of the cattail plant have been ingeniously incorporated into diverse aspects of life in various cultures throughout history. Specifically, the leaves are recognized for their strength and pliability, making them ideal for weaving into mats, baskets, and even temporary shelters, providing both comfort and protection from the elements. The Cattleya orchids in Image 2 showcase a captivating blend of pink and purple hues, accented by a vibrant yellow center. [...] | 0 (0/4) |

Table 10: Gemini response claim the task is impossible, when it's solvable

| All instructions | Question | Gemini Answer | PIF |
|---|---|---|---|
| Instruction: Only use responses to question where each sentence in the response is at most 18 words in all future responses. Instruction: Include at least one even number in the range bigger than 5 in each of your responses. Instruction: Make all the following responses at least 5 sentences. Instruction: Use the word 'per se' at least once in all future responses. Instruction: End every sentence with a question mark (?). Instruction: Start every sentence with the letter (S). | How does the structure in Image2 reflect the importance of the arts within the context of its location? | Sorry, but I cannot fulfill this request. Combining the existing instructions – to end every sentence with a question mark and to begin every sentence with the letter "S" – creates an impossible grammatical structure for a coherent response. | 0 (0/6) |

Table 11: GPT-4o ignores instructions related to numbers

| All instructions | Question | GPT-4o Answer | PIF |
|---|---|---|---|
| Instruction: Include at least one odd number bigger than 5 in each of your responses. | Describe the patterns and textures you can observe in the Image3. | In the third image, the pattern and texture are indicative of delicate lacework. The lace features intricate floral designs, with a repeated motif showcasing varying flower shapes and sizes. The detailed, openwork design creates a sense of depth and complexity, with elements of symmetry that contribute to the overall aesthetic. This type of lace pattern may require numerous stitches and precision, reflecting skilled craftsmanship. | 0 (0/6) |
| Instruction: Include at least one even number in the range bigger than 5 in each of your responses. | What activity are the individuals in the Image1 engaged in, which is characterized by rhythmic and fast-paced footwork? | In Image 1, the individuals appear to be engaged in a traditional dance involving rhythmic and fast-paced footwork, likely Tinikling, a popular folk dance from the Philippines that involves stepping deftly between bamboo poles. The dance is known for its vibrant and energetic movements, often done in groups of 2 or more participants. | 0 (0/1) |
| Instruction: Include at least one odd number bigger than 5 in each of your responses. | What type of environment is depicted in Image1? | In Image 1, the environment depicted appears to be a natural setting with tall grassy vegetation, which suggests that this environment is likely a forested or grassland area. | 0 (0/1) |
| Instruction: Include at least one odd number bigger than 5 in each of your responses. | What are some noticeable design differences between the vehicles in Image1 and Image2? | The vehicles in Image 1 and Image 2 exhibit several noticeable design differences: 1. **Body Style**: The vehicle in Image 1 is a larger, open-top car with a soft convertible roof, while the vehicle in Image 2 is a closed coupe with a hard top. 2. **Number of Doors**: Image 1's vehicle appears to be a larger model possibly with more doors, while the car in Image 2 is a 2-door coupe. 3. **Wheel Design**: The wheels on the car in Image 1 are less detailed with a vintage, simple look, but the car in Image 2 has more modern, detailed hubcaps. 4. **Fenders**: The vehicle in Image 1 has separate, rounded fenders distinct from the main body, while the vehicle in Image 2 has integrated fenders that are more streamlined with the body of the car. 5. **Overall Shape**: The car in Image 1 has a higher profile with a more upright design typical of older styles, whereas the vehicle in Image 2 has a more aerodynamic, rounded shape. These design differences highlight the evolution of automotive design over different decades. | 0 (0/1) |

