# OpenReview forum: "MMMT-IF: A Challenging Multimodal Multi-Turn Instruction Following Benchmark"
_ICLR.cc/2025/Conference — Submitted to ICLR 2025_

### Official Review · Reviewer_BesE · 2024-11-03

**Soundness:** 3
**Presentation:** 3
**Contribution:** 4
**Rating:** 6
**Confidence:** 4

**Summary:**

It introduces MMMT-IF, a challenging benchmark for evaluating multimodal, multi-turn instruction-following in AI models. This dataset extends existing multimodal benchmarks by adding global instructions that models must retrieve and follow across multiple dialogue turns, which enhances the complexity of the task. The authors propose two metrics: Programmatic Instruction Following (PIF), which measures how well models adhere to given instructions, and PIF-N-K, which tests the robustness of models by evaluating their performance across multiple response samples.

The results indicate significant challenges for models, such as Gemini 1.5 Pro, GPT-4o, and Claude 3.5 Sonnet, especially as the number of instructions or the length of the conversation increases. Human evaluations align moderately well with the PIF metric, suggesting its effectiveness in assessing instruction adherence. The study highlights that the primary difficulty lies not in following instructions per se but in retrieving and reasoning over them across long contexts.

**Strengths:**

1. MMMT-IF effectively extends the scope of existing datasets by incorporating multi-turn, multimodal challenges, making it more relevant for real-world dialogue tasks where instructions are dispersed throughout interactions.

2. The PIF and PIF-N-K metrics provide a clear and objective way to evaluate models' instruction-following capabilities, verified through code execution. This adds reliability to the evaluation process and reduces human bias.

3. The study sheds light on how current AI models struggle with retrieving and reasoning over instructions in long contexts. This insight is valuable for researchers focusing on improving the long-context reasoning capabilities of language models.

**Weaknesses:**

1. The task's high difficulty might limit its immediate usability for current models, as evidenced by significant drops in PIF scores across all tested models. This suggests that the benchmark may be too challenging for some real-world applications or standard models.

2, Implementing and evaluating models with the MMMT-IF dataset and PIF metrics can be computationally demanding, potentially hindering researchers with limited resources.

3. The need for models to follow specific and potentially complex instructions could lead to overfitting on tasks within the benchmark rather than promoting generalizable improvements in instruction-following capabilities.

**Questions:**

1. How well does the PIF metric scale with larger datasets or models? Are there limitations in applying this to more extensive, real-world data?

2. Can the MMMT-IF benchmark be adapted or simplified for single-modality tasks, or is it strictly limited to multimodal, multi-turn settings?

3. How does the complexity of individual instructions impact model performance? Would simplifying or categorizing instructions lead to better insights?

4. What are the potential improvements or training adjustments needed for models to better handle instruction retrieval and adherence?

5. How do specific architectural features (e.g., memory modules, context window size) influence model performance on the MMMT-IF benchmark?

---

> ### Author Response · Authors · 2024-11-26
> **Rebuttal by Authors**
>
> **Response to Reviewer BesE**
>
> Thank you for your thoughtful review and appreciation of our work. We respond to your comments and questions below.
>
> **W1/Q1. High Task Difficulty**
>
> The reviewer pointed to the significant drops in PIF scores as making the task highly difficult, questioning if the benchmark may currently be too challenging.
>
> We think that a task with somewhat low performance is an advantage, as there will be significant headroom for models to improve on this task, and the task may be relevant for a longer time in the future. We anticipate that models will improve on the task once multi-turn instruction following becomes more prioritized during training.
>
> **W2/Q2. Computational Cost of MMMT-IF Benchmark**
>
> The reviewer pointed out that the evaluation cost to benchmark the MMMT-IF dataset may be computationally demanding for researchers with limited resources.
>
> We provide an approximate analysis of the API cost to evaluate the OpenAI GPT-4o on our benchmark.
> We have around 1000 turns, the average input context is around 1500 tokens and the output context is on average less than 200 tokens. This gives a total number of 1.5 million input tokens and 0.2 million output tokens.
>
> For the OpenAI API, using GPT-4o, the cost is 2.5 dollars/1M input tokens, and 10 dollars/1M output tokens, hence an upper bound to evaluate GPT-4o on our benchmark is 6 dollars. We think that this would be affordable by a majority of researchers, but we also recognize that this may be limiting for some. Inherent in long context benchmarks is processing long sequences, which is costly, and this issue will be the same for other benchmarks relying on long context.
>
>
> **Q3/W3. Overfitting on Benchmark Task**
>
> The reviewer pointed out that the specific and potentially complex instructions may lead to overfitting on the benchmark tasks, that would perhaps not generalize beyond the benchmark.
>
> By having a diverse set of instruction categories, and within each category having several instructions, we hope to mitigate these issues. Even if overfitting on the specific instructions would happen, the task would still likely remain challenging as to correctly answer a question, all instructions still needs to be retrieved from the input model context.
>
> **Q4. PIF Metric Scaling to Larger Datasets/Models**
>
> The reviewer asked about how well the PIF metric scale to larger datasets/models, and asked for limitations in applying the PIF metric to real data.
>
> The computation of the PIF metric on an instruction following dataset is linear in the number of samples in the dataset.
> One limitation in applying the PIF metric to real data is that the dataset needs to contain instructions that are objective, i.e. can be accurately be checked by code execution. For example the instruction: The answer should start with the letter ‘A’ can be objectively checked by code execution, but the instruction ‘Write the answer in a positive tone’ would be hard to objectively verify by code.

---

> ### Author Response · Authors · 2024-11-26
> **Rebuttal by Authors**
>
> **Q5. MMMT-IF Adaption to Single-Modality Tasks**
>
> The reviewer asked if the MMMT-IF benchmark can be adapted to fit into single-modality tasks.
>
> The instruction following task can also work for a multi-turn task without image inputs. We include image inputs as we believe this to be an important avenue where there is a lack of instruction following benchmarks. Our approach for extending multi-turn benchmarks may be used to extend other benchmarks not using multi-modal data.
>
> One example would be to have a question answering task, but instead of basing questions on a set of images, you could base the questions on a category, such as questions about a certain country or region.
> The multi-turn aspect is important, as one of our main goals is to examine the instruction following capabilities when multiple instructions are given at different places in the input context.
>
> **Q6. Impact of Model Performance on Specific Instructions**
>
> The reviewer asked how the complexity of individual instructions impact model performance and if categorizing the instructions would lead to better insights.
>
> The complexity of the individual instructions does play a certain role in the model performance. We do categorize the instructions into 6 categories, where instructions in one category will not contradict instructions in a different category. The categories are ‘response length’ (a restriction on the number of sentences), ‘sentence start character’ (a restriction on the start character in each sentence), ‘favorite word’ (a requirement on a specific word being present in the response), ‘sentence length’ (a restriction on the length of each sentence in the response), ‘sentence end character’ (a requirement on the last character in each sentence), and ‘number’ (a requirement that the response should contain either an even or an odd number).
>
> We ensure that only one instruction per category is in each conversation. In Table 6 (page 17) of our submission we show all the given instructions. Figure 10 in the Appendix shows the PIF metric conditional on a given instruction.
>
> **Q7. Training Adjustments to Better Handle Instruction Retrieval and Adherence**
>
> The reviewer asked what are potential training adjustments needed for models to better handle instruction retrieval and adherence.
>
> This will be the focus of future work. One broad idea is to generate training data with a reward signal corresponding to the programmatic instruction following metric on responses, and then use Reinforcement Learning from Execution Feedback (RLEF) to improve the model performance on the task. This may be less costly than RLHF as we would not need to collect human preference data.
>
> **Q8. Architecture Choice Influence on the MMMT-IF Benchmark**
>
> The reviewer asked how specific architectural features influence model performance on the MMMT-IF benchmark.
>
> Context window length will matter significantly if the context window is smaller than the input context, as that would truncate the input context. In this work, we have decided to focus on uncovering an important weakness of large language models, and in future work we hope to propose different architecture adjustments and training data curation to improve performance on this task.

---

> > ### Comment · Reviewer_BesE · 2024-11-27
> >
> > Thank you for your response. The authors rebuttal have addressed my concerns. I'll maintain my positive rating.

---

### Official Review · Reviewer_GPsv · 2024-11-03

**Soundness:** 3
**Presentation:** 3
**Contribution:** 3
**Rating:** 3
**Confidence:** 3

**Summary:**

This paper introduces MMMT-IF, a multi-turn, image-based multimodal question-answering (QA) dataset designed to test models on retrieving and following instructions over extended dialogues. The benchmark challenges models to not only locate relevant instructions within long conversations but also reason effectively within the constraints of these instructions. To accurately evaluate model performance, MMMT-IF includes a metric called Programmatic Instruction Following (PIF), along with a variant, PIF-N-K, which allows for a nuanced assessment of how well models adhere to instructions. Every instruction in this dataset is verifiable through code, ensuring an objective measure of a model’s instruction-following ability. Through this benchmark, the authors demonstrate that even advanced models like Gemini-1.5-Pro and GPT-4o struggle with two key areas: reliably extracting instructions from dialogues and providing high-quality answers consistently. These challenges highlight key areas for improvement in current state-of-the-art models.

**Strengths:**

(1) This benchmark focuses on long-form multimodal dialogue QA, with conversations extending up to 20 turns. This challenging task demands the ability to understand lengthy contexts—up to 20k tokens—and helps uncover specific issues in state-of-the-art models, such as difficulties in retrieving relevant instructions or providing consistent, high-quality answers.

(2) In addition to relying on automatic evaluation, the benchmark includes human evaluation results, with a correlation score of 0.60 between human assessments and model performance, suggesting that the PIF metric is reasonably effective in measuring models’ instruction-following abilities.

(3) The benchmark also provides an in-depth analysis of potential biases in automatic evaluations, revealing that when a model acts as the evaluator, it tends to favor its own outputs. This insight helps validate the reliability of the evaluation results by accounting for potential self-assessment biases in models.

**Weaknesses:**

(1) The paper does not clearly explain how instructions are verified through code execution. Even with examples provided in the appendix, it is unclear what specific code is executed to verify each instruction. A more detailed explanation is needed, ideally with concrete examples demonstrating how instructions are translated into executable code. Additionally, the paper should define the concept of an "instruction" more clearly, as the current explanation leaves the exact meaning ambiguous.

(2) Although the paper includes detailed statistics about the dataset, it would be beneficial to include a short, complete example directly in the main text rather than in the appendix. Additionally, while the appendix briefly summarizes related work, the differences between multi-turn instruction following and related datasets like MMDU are not sufficiently explained in the main text. A clear summary of MMDU, including its main ideas and dataset characteristics, would help clarify MMMT-IF’s unique contributions. Moreover, it lacks the details of how MMDU is changed and modified to generate this proposed dataset.

(3) Expanding the evaluation to include more open-source models, such as Qwen2-VL and InternVL-2, would provide a more comprehensive assessment of model performance on the MMMT-IF benchmark. Testing open-source models also enables additional types of analysis, such as examining attention weights, which can give insights into how different models interpret and follow instructions. Currently, it only incldues analysis like needles in a haystack and sampling robustness. More in-depth analysis about the potential reason why it fails on specific cases can be done. Including a diverse set of models—spanning both interleaved and non-interleaved multimodal architectures—could reveal distinct challenges faced by different model types. While context length may pose a limitation, evaluating various open-source models could better highlight the difficulties inherent in multi-turn, multimodal instruction following, providing stronger evidence for the problem’s complexity.

**Questions:**

Could you provide more details on the human evaluation process and how the correlation with model performance is calculated? Specifically, what type of correlation metric is used (e.g., Pearson, Spearman) for these calculations? Additionally, it would be helpful to know the inter-annotator agreement statistics. While the paper indicates that human performance significantly outpaces model performance, more information on the evaluation protocol and inter-annotator correlation scores would provide a clearer understanding of the reliability and consistency of the human evaluations.

---

> ### Author Response · Authors · 2024-11-26
> **Rebuttal by Authors**
>
> **Response to Reviewer GPsv**
>
> Thank you for your thoughtful review and comments, which have made our work stronger. We respond to your comments and questions below as well as in the common response.
>
> **Q1/W1. Instruction Verification Through Code Execution**
>
> The reviewer asked for clarification on how instructions are verified through code execution and asked for a concrete example of how the instructions are checked.
>
> A chat can be represented as I1, Q1, A1, …, In, Qn, An, where Q1 is the question 1, I1 is the first instruction and A1 is the first answer.  We compute the PIF score for turn n, by using all given instructions: (I1, …, In), and An. We then execute code to first identify each instruction given, and then check if that instruction is followed in An. For instance, suppose I1= “Instruction: Use the word 'like' at least once in all future responses.”, and An = “I like ice cream”, then the following code checks if I1 is followed in A_n:
>
> ```python
> def check_word_in_answer(word, answer):
>       """ Checks if the specified word is present in the answer.
>       Args:
>          word (str): The word to look for in the answer.
>          answer (str): The response text to check. Returns: bool: True if the word is found in the answer.
>       """
>       word = word.lower()
>       answer = answer.lower()
>       return word in answer.split()
> ```
>
> We chose all instructions so that the correctness can be objectively checked by code execution.
>
> **Q2. Precise Meaning of “Instruction”**
>
> The reviewer suggested clarifying the meaning of “instruction”.
> In this paper, we define an instruction as a constraint on the answer format.
>
> **Q3. Inter-Annotator Agreement Statistics**
>
> The reviewer asked if we had numbers about the inter-annotator agreement statistics, and asked for more information on the evaluation protocol.
> Our setup has one annotator rate each dialogue, with different annotators in different dialogues. We are working on collecting multiple annotators per dialogue for a limited subset of the dataset to analyse the inter-annotator agreement in more detail.
>
> We provide the full instructions given to the annotators in appendix G of the paper. If there are some specific aspects of the evaluation protocol that are unclear, we would be happy to elaborate further.

---

### Official Review · Reviewer_TUxw · 2024-11-05

**Soundness:** 2
**Presentation:** 2
**Contribution:** 2
**Rating:** 3
**Confidence:** 4

**Summary:**

This paper introduces MMMT-IF (Multimodal Multi-Turn Instruction Following), a new benchmark for evaluating instruction following capabilities in multimodal conversational AI systems. The key contribution lies in how it extends an existing MMDU by incorporating multiple instructions that constrain response formats across conversation turns. The paper also introduces PIF metric to assess how many instructions are correctly followed during a reasoning task. The paper showed that PIF metric is aligned with human rating.

**Strengths:**

- The paper proposes two new metrics to measure the instruction following capability through program execution.
- The paper identified a significant performance drop when the number of given instructions increases.

**Weaknesses:**

- The novelty and technical contributions are somehow limited. For example, the MMMT-IF dataset is mostly an augmentation of previous MMDU dataset. The paper also lacks of providing more in-depth and insightful findings.
- Evaluated models are limited to proprietary LLMs. More open-sourced LLMs, such as LLaMA, should also be compared.
- The paper writing is not clear. For example, it would be helpful if the authors can show an example on how they extend the MMDU dataset.

**Questions:**

- It would be more interesting if the authors could provide more analysis on the model size’s effect on long-term instruction-following.

---

> ### Author Response · Authors · 2024-11-26
> **Rebuttal by Authors**
>
> ****Response to reviewer TUxw****
>
> Thank you for your review and thoughtful comments. We respond to your comments and questions below as well as in the common response.
>
>
> **Q1/W1. Concerns About MMMT-IF Being an Augmentation of MMDU**
>
> The reviewer noted that MMMT-IF is an augmentation of the MMDU dataset, suggesting that this limits the novelty of the work. We think that the setup where MMMT-IF is created based on an existing dataset is actually a strength rather than a weakness, as MMDU and MMMT-IF are testing different model capabilities, and are hence used for different tasks. MMDU is a question answering dataset, and MMMT-IF is an instruction following benchmark. Another strength with using an augmentation approach is that this augmentation strategy can be used across a range of other datasets (rather than just MMDU) to create additional datasets. Our general response talks in more depth on how the extension is made.
>
> **Q2/W2. Lack of Open-Source LLMs**
>
> The reviewer suggested that open source LLMs should be included in the benchmark.
> We thank the reviewer for this suggestion, we will include an experiment comparing with LLaMA in the camera ready version of the paper.
>
> **Q3. MMMT-IF Performance for Models of Different Size**
>
> The reviewer suggested that we examine the effect of model size on long term instruction following. For the camera ready version, we will include an experiment with different Gemini model sizes to better understand the scaling in the model size.

---

> > ### Comment · Reviewer_TUxw · 2024-12-01
> >
> > Thank you for your response. I will keep my current scores.

---

### Author Response · Authors · 2024-11-26
**Common Response**

We thank all the reviewers for their thoughtful comments, which have improved our paper.

We appreciate that the reviewers thought that the PIF metric provide a “clear and objective” (reviewer BesE) way to evaluate instruction following capabilities, that the benchmark sheds light on
models issues with “retrieving and reasoning over instructions in long contexts” (reviewer BesE),  “difficulties in retrieving relevant instructions” (reviewer GPsv), and that we show “significant performance drop when the number of given instructions increases” (reviewer TUxv). In addition, our “in-depth analysis” (reviewer GPsv) of potential biases in automatic evaluations was appreciated.
In the common response we would like to respond to questions raised by multiple reviewers.

**Differences between MMDU and MMMT-IF**

Two reviewers (reviewer TUxv, reviewer GPsv) asked us to clarify the differences between the MMDU and MMMT-IF dataset, and to describe how MMDU is extended.

MMDU is an image based question answering dataset with multiple turns. There are up to 20 turns per image pair. The questions for the same chat are all independent. The MMMT-IF dataset focuses on testing the multi-turn instruction following capabilities of LLMs, rather than focus on question answering.

An example of the extension is the following: Assume that for a pair of images IMG1, IMG2, there are questions and answers. The input context to the model at turn 5 for the MMDU benchmark would look like:

IMG1, IMG2, Q1, A1, Q2, A2, Q3, A3, Q4, A4, Q5

The A5 answer will be judged on its ability to answer question 5 correctly, and the answer metric is an LLM based judge according to the following criteria: Creativity, Richness, Visual Perception, Logical Coherence, Answer Accuracy, Image Relationship Understanding, and Overall Quality.

The MMMT-IF benchmark will use the same questions Q1, Q2, Q3, Q4, and Q5, as well as IMG1, and IMG2, but will insert instructions before questions that need to be followed for all subsequent answers.

The input context would hence be

IMG1, IMG2, I1, Q1, A1, I2, Q2, A2, I3, Q3, A3, I4, Q4, A4, I5, Q5

Here, in the answer to Q5, instructions I1, I2, I3, I4, and I5 provide constraints to the answer of question 5 that needs to be followed. To follow these, the model needs to consider the full input context, rather than just the previous question turn.

The capability we are mainly interested in here is the instruction following capabilities of the models (measured by its ability to follow constraints on the answer format, given in I1, I2, I3, I4, and I5), which is not considered for the MMDU benchmark.

Rather than single turn instruction following, our task focuses on multi-turn instruction following where instruction from several question turns needs to be followed. This is in contrast to the MMDU dataset where the focus lies in single turn question answering, as the questions are independent.

---

> ### Author Response · Authors · 2024-11-26
> **Common Response**
>
> **Example of Extension**
>
> Reviewer TUxw asked if we can show a concrete example of the extension, and reviewer GPsv asked us to include a complete example in the main text. We now show the example that we will include in the main text.
>
> Here is a short example (using Gemini 1.5 Pro for the response) that will be added to the main text in the paper:
> For turn 1 in the chat we have that
> I1 = “Include at least one even number bigger than 5 in each of your responses.”,
> Q1 = “What activity are the individuals in the Image1 engaged in, which is characterized by
> rhythmic and fast-paced footwork?“, and
> A1 = “The individuals in Image 1 are performing Tinikling, a traditional Philippine
> folk dance characterized by rhythmic and fast-paced footwork, often using
> bamboo poles. Dancers need a lot of agility to avoid getting their feet caught
> between the poles, especially as the pace increases to 10 or 12 beats per second.”
>
> In the original MMDU dataset, Q1 was the same, but it did not include the I1 instruction. At turn 15 in the same chat, all the instructions added up until that point were:
>
> “Instruction: Include at least one even
> number in the range bigger than 5 in
> each of your responses. Instruction:
> End every sentence with a exclamation
> mark (!). Instruction: Use the word
> 'itself' at least once in all future
> responses. Instruction: Only use
> responses to question where each
> sentence in the response is at most 18
> words in all future responses.
> Instruction: Make all the following
> responses no more than 4 sentences.
> Instruction: Start every sentence with
> the letter (S).”
>
> Given the input context: I1, Q1, A1, …, I15, Q15, the Gemini model produced the following answer, where Q15=”Identify the possible significance of the props or lack thereof in the settings of Image1 and Image2.”, with answer
> A15= “Stage props can enhance the narrative or symbolism of a performance. Simple
> props, like the bamboo poles in Image 1, become integral to the dance itself.
> Conversely, the absence of props in Image 2 might emphasize the dancers'
> movements and stage presence. Such choices reflect deliberate artistic
> decisions in conveying a performance's message.”
>
> Note that the answer at turn 15 needs to follow the instructions from all previous turns.

---

### Meta-Review · Area_Chair_iygE · 2024-12-20

**Metareview:**

This paper introduces MMMT-IF, a multi-turn, image-based multimodal question-answering (QA) dataset designed to test models on retrieving and following instructions over extended multi-turn dialogues. The authors also propose a metric called Programmatic Instruction Following (PIF) to assess how well models adhere to instructions. Through this benchmark, the authors demonstrate that even advanced models like Gemini-1.5-Pro and GPT-4o struggle to extract instructions from dialogues and provide high-quality answers consistently.

2 reviewers were negative about the paper, and one gave marginal above acceptance. The reviewers felt that the proposed dataset is interesting and presents a challenge to today's latest models, especially wrt the multi-turn aspect, and proposed metrics that can be verified through code execution.

However, they found the novelty limited since it is largely an extension of the existing MMDU dataset, there were issues with the clarity of writing, the benchmark may be too challenging, and reviewer GPsv also raised issues with the depth of experimental analysis including missing evaluation of open-source models and in-depth analysis of failure cases. I am aligned with the majority of the reviewers' judgement that the benchmark is a good step, but there are issues with the motivation, writing, and analysis, so the paper falls below the threshold for acceptance.

**Additional Comments On Reviewer Discussion:**

Reviewer TUxw and reviewer BesE acknowledged the author feedback but stayed with their negative and positive scores respectively. Reviewer GPsv raised several important weaknesses, including improving the writing and presenting more concrete examples and dataset characteristics, and details of how MMDU is changed and modified to generate this proposed dataset. They also suggested expanding the evaluation to include more open-source models, such as Qwen2-VL and InternVL-2 to provide a more comprehensive assessment of model performance, more in-depth analysis of specific failure cases, etc. The authors did not respond adequately to these concerns and in particular the point about providing in-depth analysis with the recent public models and analysis of failure cases, so I would consider them unaddressed. In light of these issues, I believe the paper falls below the threshold for acceptance.

---

### Decision · Program_Chairs · 2025-01-22

Reject